# Is an Increased Risk of Developing Guillain–Barré Syndrome Associated with Seasonal Influenza Vaccination? A Systematic Review and Meta-Analysis

**DOI:** 10.3390/vaccines8020150

**Published:** 2020-03-27

**Authors:** Marek Petráš, Ivana Králová Lesná, Jana Dáňová, Alexander M. Čelko

**Affiliations:** 1Department of Epidemiology and Biostatistics, Charles University in Prague-Third Faculty of Medicine, 100 00 Prague, Czech Republic; jana.danova@lf3.cuni.cz (J.D.); martin.celko@lf3.cuni.cz (A.M.Č.); 2Laboratory for Atherosclerosis Research, Centre for Experimental Medicine, Institute for Clinical and Experimental Medicine, 140 21 Prague, Czech Republic; ivka@ikem.cz

**Keywords:** Guillain–Barré syndrome, vaccine against seasonal influenza, influenza-like illness, vaccine-associated autoimmune disease

## Abstract

While the weight of epidemiological evidence does not support a causal link with influenza vaccination evaluated over the last 30 years, Guillain–Barré syndrome (GBS) has been considered a vaccine-associated adverse event of interest since 1976. To investigate the existence of GBS risk after vaccination against seasonal influenza, a systematic review and meta-analysis have been conducted based on 22 eligible epidemiological studies from 1981 to 2019 reporting 26 effect sizes (ESs) in different influenza seasons. The primary result of our meta-analysis pointed to no risk of vaccine-associated GBS, as documented by a pooled ES of 1.15 (95% CI: 0.97–1.35). Conversely, an obvious high risk of GBS was observed in patients with previous influenza-like illness (ILI), as demonstrated by a pooled ES of 9.6 (95% CI: 4.0–23.0) resulting from a supplementary analysis. While the meta-analysis did not confirm the putative risk of vaccine-associated GBS suggested by many epidemiological studies, vaccination against seasonal influenza reduced the risk of developing ILI-associated GBS by about 88%. However, to obtain strong evidence, more epidemiological studies are warranted to establish a possible coincidence between vaccination and ILI prior to GBS onset.

## 1. Introduction

Guillain–Barré syndrome (GBS) is a rare but serious cause of acute neuromuscular paralysis, resulting in muscle weakness and a loss of reflexes in about two cases per 100,000 people per year [1,2]. The syndrome has been shown to be associated with antecedent gastrointestinal or upper respiratory tract infections, including influenza [3,4,5]. Although the exact causes of GBS are unknown, an individual’s genetic background might enable the generation of autoimmune antibodies triggered by either molecular mimicry or bystander activation, resulting in the demyelination of and damage to peripheral nerves [6,7].

The initiative of rapid vaccination of the USA population against swine influenza in 1976 entailed an unexpected increase in GBS risk in adults, alerting researchers to a dangerous vaccine association with GBS [8,9,10]. Since then, GBS has been a potential vaccine-associated adverse event of interest, although later studies looking into the association between GBS and influenza vaccination have suggested its absence [11,12,13,14,15,16,17,18,19,20,21,22]. However, a possible causal link was again documented in summary studies designed to evaluate GBS onset after immunization with a monovalent vaccine only used in the 2009–2010 season to prevent a pandemic of influenza type A subtype H1N1 [23,24,25]. 

Therefore, we conducted a meta-analysis in which all epidemiological studies available from our computerized search were entered to develop a risk estimate of influenza vaccine-associated GBS. We only focused on GBS occurrence after immunization against seasonal influenza carried out every year using a common inactivated trivalent vaccine, in order to exclude the effect of the extraordinary use of a monovalent pandemic influenza vaccine with or without adjuvants. We aimed to evaluate the effect size (ES) not affected by potential confounders resulting from the new pandemic strain of influenza A subtype H1N1, rapid manufacture and distribution of a pandemic influenza vaccine, etc.

## 2. Materials and Methods

This systematic review was based on the guidelines outlined in the Preferred Reporting Items for Systematic Reviews and Meta- analyses (PRISMA) Statement [26], using the Medical Literature Analysis and Retrieval System Online (MEDLINE), Excerpta Medica dataBASE (EMBASE), Derwent Drug File (DDFU), ProQuest Science & Technology (POSCITECH), BioSciences Information Service of Biological Abstracts (BIOSIS), and Chemical Abstracts Plus (HCAPLUS) databases from the earliest date available through 30 November 2019, with no language restrictions. The key words used were vaccine or vaccination, in combination with flu or influenza and Guillain–Barré syndrome, including their synonyms (see Appendix A). Moreover, a recursive search of reference lists of full-text journals was conducted to identify articles not included in our computerized search. 

Eligible studies had to meet the following selection criteria: (1) observational studies with controls such as case-control (C-C) and cohort (C) studies, including their modifications, i.e., self-controlled case-series (SCCS) or self-controlled risk interval (SCRI), as well as case cross-over (CCo) and case-centered (Cc) studies; (2) exposure to vaccination against seasonal influenza; (3) control group without vaccination; and (4) outcome of interest measured using the ES of vaccination on GBS occurrence, including the 95% confidence interval (excluding studies without reported ES or entry for odds ratio calculation).

Data were independently extracted by two reviewers from eligible publications using a predefined format. If consensus was not achieved, the discrepancy was resolved by discussions among all authors. Extracted data contained the authors’ names; year of publication; study period or influenza season; country; participants, interventions, comparisons, outcomes, and study design (PICOS); time window (cases reported during the at-risk interval after immunization); ES used; and GBS risk estimate with 95% CI, factors of adjustment, or assessment.

The primary result of the pooled ES was obtained from the general population, irrespective of sex and age, where only study outcomes of the longest time windows were taken into account.

The studies’ quality was assessed according to the Newcastle–Ottawa Scale (NOS) in three domains: selection of participants, comparability of vaccinated and unvaccinated participants, and outcome of interest [27]. The total NOS score of 9 stars was divided into three categories of study bias: low bias (7–9), moderate bias (4–6), and high bias (0–3). Studies with an NOS score ≥ 7 stars were entered as a set of high-quality studies used to determine the primary outcome of pooled ES. The robustness of the primary outcome was tested in a set of all studies. 

Study-reported relative risks, incidence rate ratios, or odds ratios with or without adjustment were used for a pooled quantitative analysis, i.e., for pooled ESs. The risk of developing GBS associated with vaccination against seasonal influenza was estimated from the pooled ES with a 95% confidence interval CI pursuant to a random-effect model (DerSimonian–Laird method; D-L), because the heterogeneity of studies’ outcomes did not allow us to employ a fixed-effect model (inverse variance method, I–V).

Furthermore, the results of primary analysis were tested against the strength of evidence using the Grading of Recommendations Assessment, Development and Evaluation (GRADE) system [28] with the following criteria: (1) a sufficient number of studies (≥10) [29]; (2) no serious limitations of studies ensured by their selection with an NOS score ≥ 7 stars; (3) no serious inconsistency ensured by the studies’ heterogeneity I^2^ < 75% [30]; (4) no serious indirectness ensured by high-quality studies with complete comparability of both study groups; (5) no serious imprecision of studies with a standard error (SE) < 0.1 of pooled ES, i.e., ±10% error of outcome; and (6) no serious publication bias if both ESs, as determined by fixed and random effects, provided the same association [31]. Moreover, the presence of a small study effect and the risk of publication bias were tested as follows: (a) the effect of small studies was determined by the regression model with Egger’s test, and (b) the summary effect of asymmetry while identifying any unpublished studies was estimated by the trim-and-fill method. The robustness of pooled ES was investigated in the set of all studies, irrespective of their quality and if the outcome provided the same association, the primary outcome was considered to be robust.

Statistical analyses were performed using STATA version 15.1 (StataCorp. 2017. Stata Statistical Software: Release 15. College Station, TX, USA). 

## 3. Results

### 3.1. Selection and Characteristics of Studies

A total of 421 publications were identified (Figure 1), of which number 343 were discarded after reading their titles and abstracts due to overlapping or obviously irrelevant studies. The 22 eligible publications which met the inclusion criteria were published between 1981 and 2019. Since some studies provided more than one outcome of interest, their number, eventually pooled in the quantitative analysis, increased to 26 study entries (Table 1).

The development of GBS was documented using the International Classification of Diseases, Ninth Revision, code 357.0 (n = 10) or confirmed according to the Brighton Collaboration definition of levels 1–3 and/or possibly 4 (n = 9). In two studies, GBS was diagnosed by neurologists of the Committee of the National Institute of Neurologic and Communicative Disorders and Stroke or using inclusion criteria similar to the Brighton Collaboration definition. One study failed to specify GBS ascertainment at all.

A total of 17 studies evaluated the risk of GBS after immunization with an inactivated trivalent vaccine against seasonal influenza (iTIV). Two studies failed to provide a specification of the influenza vaccine, but use of the inactivated vaccine was highly likely in the studies’ geographic region (China). Although the live influenza vaccine has been used in the USA since 2003, only a negligible number of participants were immunized with it, as reported by three studies.

The association between vaccination and GBS risk was investigated in six case-control studies and in five cohort, self-controlled case-series and self-controlled risk interval studies each, plus one case-centered study. A total of 11 studies were based on a time window. Twelve studies were performed in North America (10 in the USA and two in Canada) and five in Europe (three in the UK, one in France, and one in Italy). The others were conducted in Asia (China and Taiwan) or Australia. The investigated association was most often observed during influenza seasons in the 2009-11 period (six studies). A total of 15 studies assessed this risk within 42 days (6 weeks) of vaccination. Eleven studies were designed with longer time windows. Only two studies [33,34] evaluated risk estimates separately for males and females, which was why no sensitivity analyses of sex were performed. 

A total of 16 studies focused on GBS risk estimates in the general population, independent of age. An age-specific association was evaluated in 10 studies, but the non-uniform age distribution across them did not make it possible to quantitatively assess the pooled ES for age-specific groups, except for people ≥ 65 years of age investigated in four studies.

To prevent a high risk of bias, the primary analysis was conducted on a set of high-quality studies, i.e., 18 of them met an NOS score ≥ 7 stars. Another three had a score of 6, while one achieved a score of 5 (see Appendix A).

### 3.2. iTIV Vaccination and GBS Risk

The pooled risk estimate of developing GBS obtained from our primary analysis did not show any significant increase after immunization against seasonal influenza, i.e., ES = 1.15 (95% CI: 0.97–1.35) (see Figure 2). The pooled outcome was not burdened by a serious limitation and indirectness of studies, since the analysis was only conducted on selected individuals with an NOS score ≥ 7. Moreover, their inconsistency (I^2^ = 40.5%) and imprecision (SE = 0.08) were not serious. 

The absence of an increased risk of developing GBS after iTIV immunization was also confirmed by a set of all studies, independent of their quality, i.e., ES = 1.15 (95% CI: 0.99–1.35), so the pooled GBS risk estimate could be considered robust. 

Additional tests did not find any effect of small studies, asymmetry of entries, or the absence of unpublished studies. However, the risk of publication bias did exist, since there were different associations determined by ESs of both fixed- and random-effect models that could eventually influence the pooled outcome. The mere presence of a bias risk did not permit us to achieve the required strength of evidence. 

### 3.3. Estimate of GBS Risk in Specific Subgroups

The impact of specific conditions on the outcome of interest was tested using sensitive meta-analyses of subsets of studies with different time windows, age-specific groups, methodologies, and geographic regions or influenza seasons. The pooled outcomes were obtained from at least three studies and are displayed in Table 2.

Our sensitivity analysis tested the impact of shorter (i.e., interval from 1st day to 42th day) and longer (i.e., interval from 1st day to 43rd–365th day) post-vaccination time windows on the GBS risk estimate. The pooled outcomes for shorter and longer time windows did not demonstrate any association between GBS occurrence and iTIV vaccination, i.e., ES = 1.19 (95% CI: 0.99–1.44) and ES = 1.08 (95% CI: 0.77–1.52), respectively.

The risk estimate in age-specific groups was only limited to a population over 65 years of age, where no increased risk of developing GBS was found, as documented by the ES value of 1.11 (95% CI: 0.88–1.39).

Except for cohort studies, all the methodologies used provided consistent outcomes showing no vaccine-associated GBS. The pooled outcome obtained from cohort studies only suggested an increase in GBS risk, i.e., ES = 1.56 (95% CI: 1.16–2.09), *p* = 0.003.

While the pooled outcome of studies conducted in Europe confirmed no post-vaccination GBS risk, the pooled outcome of North American studies did exhibit an increased risk, i.e., ES = 1.24 (95% CI: 1.02–1.50), *p* = 0.026.

The analysed results of studies focused on the influenza season of 2009–2010 (i.e., the season of the influenza pandemic), which provided an ES proving an association between GBS and vaccination, i.e., ES = 1.43 (95% CI: 1.03–1.97), *p* = 0.032. Conversely, this association was not observed in the following season (2010–2011), i.e., ES = 1.35 (95% CI: 0.88–2.07), *p* = 0.179.

The pooled risk estimates of post-vaccination GBS obtained from sensitivity analyses could not be generalized because all of them were burdened by an unacceptable imprecision (SE ≥ 0.10). In addition, the sample size of subsets was usually insufficient (n < 10). 

### 3.4. Influenza and GBS Risk

A total of five studies also investigated the possibility of an association between Guillain–Barré syndrome onset within 30 to 180 days of influenza-like illness (ILI) or upper respiratory infection. The pooled ES from these studies—ES = 9.6 (95% CI: 4.0–23.0)—showed a highly increased risk of GBS (Figure 3). The same outcome was observed in both fixed- and random-effect models. While the association was not burdened by the publication bias of studies, the studies exhibited an unacceptable heterogeneity (I^2^ = 90.5%).

The GBS risk was significantly lower in patients with previous immunization than in those with previous ILI, as confirmed by the ES ratio, i.e., ES_iTIV_/ES_ILI_ = 0.12 (95% CI: 0.05–0.29). Compared to ILI, vaccination against seasonal influenza decreased the risk of GBS onset by about 88% (95% CI: 71–95%).

## 4. Discussion

This meta-analysis did not reveal an increased risk of developing GBS any time after vaccination against seasonal influenza, but a secondary outcome pointed to an undisputable risk of GBS in patients with previous ILI. In spite of the incomplete strength of evidence, the pooled ES of 1.15 (95% CI: 0.97–1.35) was clearly very similar to the results of another meta-analysis published in 2015, i.e., a marginally vaccine-associated GBS risk of 1.22 (95% CI: 1.01–1.48) [25].

Unlike a previous meta-analysis, ours included an additional four studies and evaluated a total of 26 study-entered ESs. A more accurate outcome was achieved using the following strategy: (1) bias risk was limited by a rigorous assessment of the study quality; (2) effect sizes of interest were derived from high-quality studies only; and (3) estimation of standard errors ensured an original association of published studies.

The outcomes of sensitivity analyses confirmed GBS risk, independent of the short- or long-time post-vaccination interval. Even if some studies documented, in time windows of ≤42 days [37,38] or ≤49 days [34], a vaccine-associated increase in GBS, the pooled risk estimate within 42 and 43-365 days de facto did not demonstrate this marginal association.

It is not likely that the risk estimate of GBS is influenced by age-specific populations. Similarly, an increased risk was not documented by the pooled ES in people over 65 years, as well as by study-determined ESs in children and adolescents under 18 years [33,38] or people younger or older than 25 years [17,36]. However, when scrutinizing the risk estimate in the population aged ≥ 18 years [34,39] or between 18 and 64 years [32,38], the association outcomes were not consistent, thus implying an unproven causality of the vaccine.

The pooled risk estimate in subgroups in various types of studies, except for cohort ones, did not reveal any association between GBS and vaccination. Although the ES reported by cohort studies did not demonstrate any causality, the pooled ES showed a potential vaccine-associated GBS onset. At the same time, this type of methodology did not seem to be more sensitive than the others. The explanation for this may lie in the insufficient number of required entries, intensified by the estimation approach of quantitative analyses.

The causality found in the 2009–2010 season resulting from the pooled outcome could be explained in the same way, because no vaccine-associated GBS for that season was documented in several studies [17,18,36,39]. Likewise, no increased risk of post-vaccination GBS was reported for the next season of 2010-11. 

While the studies conducted in European countries did not find any risk, a somewhat higher risk of developing GBS was documented by the pooled ES in North American studies. Two Canadian studies [34,38] with increased risk estimates of post-vaccination GBS contributed to this outcome, where a coincidence of previous ILI and influenza vaccination in the risk interval was documented in their patients.

Our review of publications has identified other factors that have an impact on the increased risk of GBS. A total of five studies showed a strong association of GBS with influenza, ILI, or upper respiratory infections, commonly referred to as influenza-like illness under the umbrella term [4,14,16,37,38]. Therefore, our quantitative analysis was undertaken to assess the pooled risk estimate of GBS in patients with previous ILI. Comparing the pooled ESs of both iTIV and ILI, an 88% decrease in GBS was observed in vaccinated persons against influenza-ill patients.

Furthermore, some studies have reported ILI in vaccinated patients within the GBS risk interval [37,38]. It is clear that immunization against influenza can ensure the sufficient protection of about 50%–80% vaccinees within the first 2–4 weeks. A limitation of vaccine-elicited protection results from a potential deviation between circulating and vaccination strains of influenza viruses. In addition, the vaccination cannot obviously protect against other viral or bacterial respiratory illnesses. If taking into account the onset of influenza and other respiratory infections typically occurring between October and December, and coinciding with seasonal vaccination, the potential risk of GBS cannot be attributed to immunization only. A similar finding was noted in a study where 67 of 69 vaccinated GBS patients had contracted influenza during the risk interval [38]. It is likely that some study-reported risk estimates of GBS in vaccinated patients could reflect vaccine failure or coincident respiratory infection within the influenza season. 

Admittedly, not all published references were identified in this systematic review and meta-analysis, albeit no additional references being retrieved from the reference lists in the articles emerging from the literature search, which we accept as a limitation of our study.

## 5. Conclusions

The outcome of the present study points to no increase in the risk of GBS onset in patients who have previously received an inactivated trivalent vaccine against seasonal influenza, irrespective of the post-vaccination time. While the weight of epidemiological evidence does not support a causal link, the absence of a vaccine-associated GBS risk resulting from this analysis must be considered with caution because the strength of evidence has not been achieved. The benefit of influenza vaccination was shown by the 88% decrease in Guillain–Barré syndrome occurrence compared to patients with previous ILI.

Therefore, it would be desirable to conduct other studies where the coincidence of vaccination and respiratory infection diseases is separated. This could help to better investigate the real vaccine-related impact on Guillain–Barré syndrome and to dispel any potential doubts about vaccination against seasonal influenza. 

## Figures and Tables

**Figure 1 vaccines-08-00150-f001:**
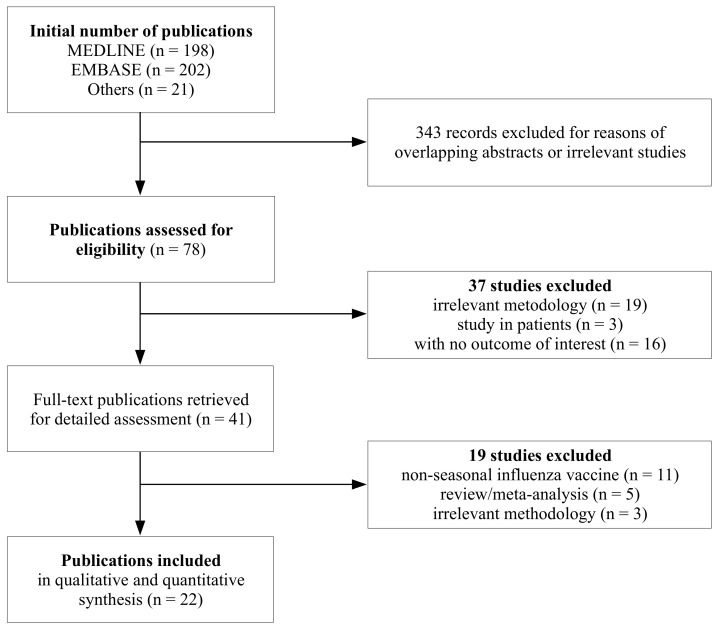
Flowchart.

**Figure 2 vaccines-08-00150-f002:**
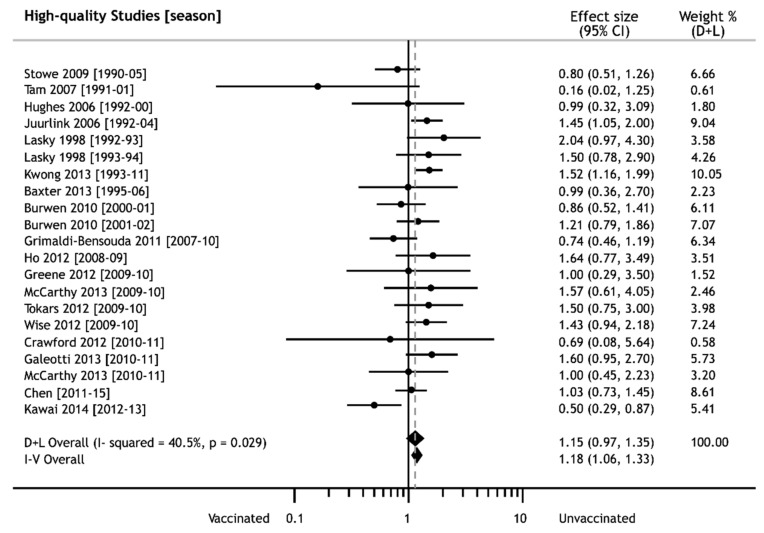
Forest plot of effect sizes of Guillain–Barré syndrome after seasonal influenza vaccination; weights from random effects analysis (dashed line in point of pooled ES).

**Figure 3 vaccines-08-00150-f003:**
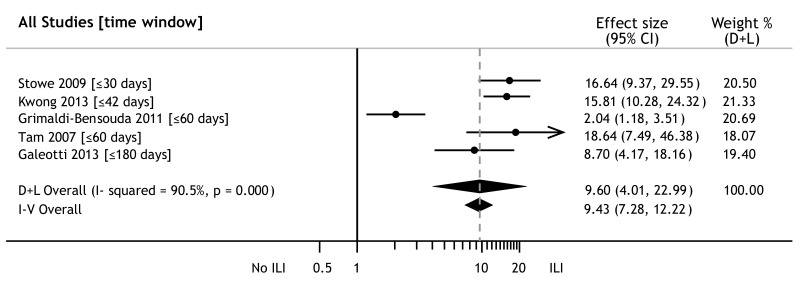
Forest plot of effect sizes of Guillain–Barré syndrome after influenza-like illness; weights from random effects analysis.

**Table 1 vaccines-08-00150-t001:** Characteristics of 22 observational studies included in the meta-analysis.

Study, Year [Ref.]; Country; Study Period	Study Design	Participants	iTIV^1^ (%)	GBS^2^ Assessment	Time Window (Days)	Effect Size (95% CI^3^)	Quality
Hurwitz 1981 [11]; USA; 1978–1979	C^4^	F^9^: 44%; Age: 0–95 y^10^; 544 pts^11^	100	Committee^12^	≤56	cRR^13^:1.4 (0.7–2.7)	N^14^
Kaplan 1982 [12]; USA; 1979–1981	C	F: 43%; Age: 15–74 y; 778 pts	100	Neurologists	≤56	cRR:0.6 (0.29–1.31) S_1979–1980_^21^cRR:1.4 (0.8–2.5) S_1980–1981_	N
Lasky 1998 [32]; USA; 1992–1994	C	F: 38%; Age: ≥18 y; 273 pts	100	ICD-9: 357.0^15^	≤42	aRR^13^:2.0 (<1.0–4.3) S_1992–1993_aRR:1.5 (0.8–2.9) S_1993–1994_	Y^14^
Liu 2003 [33]; China; ?	C-C^5^	F: 49%; Age: 1–14 y; 51 pts and 51 controls	100^16^	Selection criteria^17^	Any time	OR^18^:3.13 (0.27–82.33)	N
Hughes 2006 [13]; UK; 1992–2000	SCCS^6^	F: 47%; Age: 0+; 228 pts	100	ICD-9: 357.0	≤42	aRR:0.99 (0.32–3.12)	Y
Juurlink 2006 [34]; Canada; 1992–2004	SCCS	F: ?^19^; Age: ≥18 y; 685 pts	100	ICD-9: 357.0	≤49	aRR:1.45 (1.05–1.99)	Y
Tam 2007 [4]; UK; 1990–2001	C-C	F: ?; Age: 0+ y; 553 pts and 5445 controls	100	ICD-9: 357.0	≤60	aOR^18^:0.16 (0.02–1.25)	Y
Stowe 2009 [14]; UK; 1990–2005	SCCS	F: 43%; Age: 0+ y; 690 pts	100	ICD-9: 357.0	≤30≤180	aRR^20^:0.58 (0.18–1.86)aRR:0.80 (0.51–1.27)	Y
Burwen 2010 [15]; USA; 2000–2002	SCRI^7^	F: 60%; Age: 0+ y; 164 pts	100	BCC: 1,2^22^	≤42	IRR^20^:0.86 (0.52–1.41) S_2000–2001_IRR:1.21 (0.79–1.86) S_2001–2002_	Y
Grimaldi-Bensouda 2011 [16] France; 2007–2010	C-C	F: 39%; Age: 3–80 y; 145 pts and 1080 controls	100	BCC: 1,2,3	≤42≤180	aOR:1.3 (0.41–4.12)cOR4:0.74 (0.46–1.19)^23^	Y
Ho 2012 [35]; Taiwan; 2008–2009	C	F: 52%; Age: ≥65 y; 41,986 vaccinated and 51,063 unvaccinated	100	ICD-9: 357.0	≤365	aOR:1.64 (0.77–3.49)	Y
Tokars 2012 [36]; USA; 2009-2010	SCRI	F: 58%; Age: 2–88 y; 78 pts	>51,3	BCC: 1,2,3	≤42	RR:1.5 (0.8–3.0)^24^	Y
Wise 2012 [17]; USA; 2009–10	C	F: 48%; Age: 0+ y; 411 pts	>53	BCC: 1,2,3	≤42	aRR:1.43 (0.94–1.89)	Y
Greene 2012 [18]; USA; 2009–2010	SCRI	F: 63%; Age: 2–83 y; 14 pts	100	BCC: 1,2,3	≤42	RR:1.0 (0.3–3.5)	Y
Crawford 2012 [19]; Australia; 2010–2011	SCCS	F: 48%; Age: 7–95 y; 54 pts	100	BCC: 1,2,3,4	≤42	IRR:0.69 (0.08–5.64)	Y
Baxter 2013 [20]; USA; 1995–2006	Cc^8^	F: 41%; Age: 5–87 y; 451 pts	100	BCC: 1,2,3	≤42	aOR:1.11 (0.39–3.08)	Y
					≤70	aOR:0.99 (0.33–2.70)	
Galeotti 2013 [37]; Italy; 2010–2011	C-C	F: 42%; Age: ≥18 y; 140 pts and 308 controls	100	BCC: 1,2,3	≤42	aOR:3.8 (1.3–10.5)	Y
					≤365	aOR:1.6 (0.9–2.7)	
Kwong 2013 [38]; Canada; 1993–2011	SCCS	F: 46%; Age: 0+ y; 330 pts	100	ICD-9: 357.0	≤42	IRR:1.52 (1.17–1.99)	Y
McCarthy 2013 [39]; USA; 2009–2011	SCRI	F: ?; Age: 0–80 y; 1021 pts	?	ICD-9: 357.0	≤42	aRR:1.57 (0.61–4.05) S_2009–2010_aRR:1.00 (0.45–2.23) S_2010-2011_	Y
Kawai 2014 [21]; USA; 2012–2013	SCRI	F: 44%; Age: 0+ y; 116 pts	100	ICD-9: 357.0	≤42	aRR:0.5 (0.3–0.9)	Y
Chang 2019 [22]; Taiwan; 2007–2015	C-C	F: 38%; Age: ≥50 y; 182 pts and 910 controls	100	ICD-9: 357.0	≤42≤90	OR:1.46 (0.56–3.78)OR:1.26 (0.67–2.38)	Y
Chen 2019 [40]; China; 2011–2015	C-C	F: 38%; Age: 0+ y; 1056 pts and 4312 controls	100^16^	BCC: 1,2,3	≤42	aOR:1.03 (0.73–1.45)	Y

^1^ iTIV, inactivated trivalent influenza vaccine; ^2^ GBS, Guillain–Barré syndrome; ^3^ CI, confidence interval; ^4^ C, cohort study; ^5^ C-C, case-control study; ^6^ SCCS, self-controlled case-series; ^7^ SCRI, self-controlled risk interval; ^8^ Cc, case-centered studies; ^9^ F, female; ^10^ y, years; ^11^ pts, patients; ^12^ Committee of National Institute of Neurologic and Communicative Disorders and Stroke; ^13^ cRR/aRR, crude/adjusted relative risk; ^14^ N/Y, no/yes; ^15^ high probability of iTIV; ^17^ selection criteria corresponding to BCC 1,2; ^18^ OR/aOR, crude/adjusted odds ratio; ^19^ ?, not reported; ^20^ IRR/aIRR, crude/adjusted incidence rate ratio; ^21^ S, influenza season; ^22^ BCC, Brighton Collaboration Criteria; ^23^ calculated from study data; ^24^ fixed-window method considered more accurate by the authors, with less potential bias from seasonal effects.

**Table 2 vaccines-08-00150-t002:** Risk estimate of Guillain–Barré syndrome expressed by the pooled effect size in specific subgroups (sensitivity analyses).

Critical Variable		High-Quality Studies		All Studies	
		Study Records	ES^1^ (95% CI^2^)	Study Records	ES (95% CI)
Time window	shorter^3^	17	1.19 (0.99–1.44)	18	1.20 (1.00–1.44)
	longer^4^	7	1.08 (0.77–1.52)	12	1.12 (0.89–1.42)
Age (years)	≥65	5	1.11 (0.88–1.39)	5	1.11 (0.88–1.39)
Study (type)	C-C	4	0.97 (0.62–1.53)	6	1.05 (0.74–1.48)
	C	4	1.56 (1.16–2.09)	7	1.39 (1.10–1.75)
	SCCS	5	1.25 (0.94–1.65)	5	1.25 (0.94–1.65)
	SCRI	7	0.98 (0.72–1.34)	7	0.98 (0.72–1.34)
Geographic region	North America	13	1.24 (1.02–1.50)	16	1.22 (1.02–1.45)
	Europe	5	0.90 (0.58–1.40)	5	0.90 (0.58–1.40)
Influenza season	2009–2010	4	1.43 (1.03–1.97)	4	1.43 (1.03–1.97)
	2010–2011	3	1.35 (0.88–2.07)	3	1.35 (0.88–2.07)

^1^ ES, effect size determined using the random-effect model; ^2^ CI, confidence interval; ^3^ shorter time window (from 1st day to 42th day); ^4^ longer time window (from 1st day to 43rd–365th day).

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
