# Peer review of "Is an Increased Risk of Developing Guillain–Barré Syndrome Associated with Seasonal Influenza Vaccination? A Systematic Review and Meta-Analysis"

_vaccines, 2020, doi:10.3390/vaccines8020150_

Round 1

Reviewer 1 Report

The manuscript entitled "Is increased risk of developing Guillain-Barre syndrome associated with seasonal influenza vaccination? A systematic review and meta-analysis," by Petras et al. describes the meta-analysis of results of 22 studies to investigate whether vaccination for seasonal influenza is associated with an increased risk of developing Guillain-Barre syndrome. Overall, the study is well written, and the results are well presented. There are number of spelling errors that should be addressed prior to publication (e.g. Figure 1: irrelevant metodology, Table 1: yeas/no [also please reorder: N/Y … no/yes] etc.). One issue that should be taken in consideration by the authors is the interpretation of effect size regarding association. For instance, the authors state that “…provided an ES proving an association between GBS and vaccination, i.e. ES=1.43. Conversely, this association was not observed in the following season, i.e. ES=1.35.” Assuming normal distribution, the difference in CLES between these effect sizes is roughly only 0.02. It is unclear why one was interpreted as showing an association while the other was not.

Author Response

Responses to comments of Reviewer No 1

First, I would like to thank you for your valuable comments/suggestions.

The manuscript entitled "Is increased risk of developing Guillain-Barre syndrome associated with seasonal influenza vaccination? A systematic review and meta-analysis," by Petras et al. describes the meta-analysis of results of 22 studies to investigate whether vaccination for seasonal influenza is associated with an increased risk of developing Guillain-Barre syndrome. Overall, the study is well written, and the results are well presented. There are number of spelling errors that should be addressed prior to publication (e.g. Figure 1: irrelevant metodology, Table 1: yeas/no [also please reorder: N/Y … no/yes] etc.).

Answer:

Thank you for your remark, the typos have been corrected.

Figure 1

irrelevant metodology was changed to “irrelevant methodology”

Table 1

N/Y … yeas/no was changed to N/Y … no/yes

One issue that should be taken in consideration by the authors is the interpretation of effect size regarding association. For instance, the authors state that “…provided an ES proving an association between GBS and vaccination, i.e. ES=1.43. Conversely, this association was not observed in the following season, i.e. ES=1.35.” Assuming normal distribution, the difference in CLES between these effect sizes is roughly only 0.02. It is unclear why one was interpreted as showing an association while the other was not.

Answer:

The association was confirmed by 95% CI, i.e. if this interval contains a value of 1, then the association was not proven. Otherwise a positive or a negative association is found. I agree that it would be better to report p-value.

Completion/Correction (lines 233-242):

Analysed results of studies focused on the influenza season of 2009-10 (i.e., the season of the influenza pandemic) provided an ES proving an association between GBS and vaccination, i.e. ES = 1.43 (95% CI: 1.03-1.97), p=0.032. Conversely, this association was not observed in the following season (2010-11), i.e. ES = 1.35 (95% CI: 0.88-2.07), p=0.179.

Reviewer 2 Report

This systematic review and meta-analysis (SRMA) is an updated version of paper by Martin Arias et al. published in 2015. Although the topic is of high interest for the “Vaccines” the reporting quality is poor.

General comments:

  • The paper must be reviewed by a native English speaker;
  • The paper must follow the PRISMA guidelines.

Introduction

  • The study rationale should be described better. Given that you are updating an existing SRMA, you should describe the main results of the SRMA by Martin Arias et al. Moreover, the same research group updated their SRMA in 2019, by identifying further three studies (Sanz Fadrique R, Martín Arias L, Molina-Guarneros JA, et al. Guillain-Barré syndrome and influenza vaccines: current evidence. Rev Esp Quimioter. 2019 Aug;32(4):288-295.). The latter paper was not cited here. Why it is important to update the existing SRMA?
  • The phrase “Therefore, we conducted this meta-analysis where all currently available epidemiologic studies were entered to develop a risk estimate of influenza vaccine-associated GBS” is misleading. You cannot know whether you included all available studies. For instance, the following studies are missing: 1. Sandhu SK, Hua W, MaCurdy TE, Franks RL, Avagyan A, Kelman J, et al. Near real-time surveillance for Guillain-Barre syndrome after influenza vaccination among the Medicare population, 2010/11 to 2013/14. Vaccine. 2017;35(22):2986-92; 2. Ghaderi S, Gunnes N, Bakken IJ, Magnus P, Trogstad L, Haberg SE. Risk of Guillain-Barre syndrome after exposure to pandemic influenza A(H1N1)pdm09 vaccination or infection: a Norwegian population-based cohort study. Eur J Epidemiol. 2016;31(1):67-72.
  • The phrase “To exclude potential risk or confounder factors such as “experimental” influenza vaccine, unpredictable circulating strain or extraordinary vaccination, we focused only on GBS occurrence after immunisation against seasonal influenza carried out every year using a common inactivated trivalent vaccine” has several issues. First, what do you mean by “experimental” vaccines? You included only observational studies; this means that only licensed products could be investigated in primary research. Second, you cannot exclude the confounding by “unpredictable circulating strains”: both the virus population and vaccine composition change almost each season. In order to minimize such an effect you have to pool the available studies separately by season and location and/or to perform a meta-regression analysis. By the term ”extraordinary” vaccination do you intend the pandemic one? What about quadrivalent vaccines? They have been the most widely used vaccines for the recent 2-3 years (in several high-income countries).

Methods

  • Do you have a protocol? Have you registered it. If yes, you should provide a reference to it or at least attach it in a supplementary file.
  • You have to report at least one search syntax in full.
  • Did you search the grey literature? These sources are of interest here (e.g. VAERS-based evaluations).
  • The inclusion criterion 1 has to be reviewed. For instance, the cohort design does not have cases and controls.
  • Why the randomized controlled studies were excluded? Sufficiently powered RCTs (usually efficacy trials) have thousands of study subjects. You may have a rate of GBS in both vaccine and placebo/non-influenza vaccines arms.
  • Please specify the exclusion criteria. The phrase “Studies without usable data were excluded” is not sufficient (it is also “unclear”).
  • Please specify which effect sizes you have considered a priori.
  • The PICOS algorithm must be specified. For instance, which population? How did you categorize the population?
  • Factors of adjustment were not reported in the results.
  • Do you have a reference for the NOS scale score categorization? As far as I know currently there is no a well-defined threshold for distinguishing between high and low quality studies.

Results

  • The list of excluded studies should be reported in a supplementary file.
  • Is this a typo ≤43 and ≤43-365 days?
  • No results from the declared publication bias analysis were reported.
  • No results of the GRADE evaluation were reported.
  • You pooled together different study designs and associated different effect sizes like risk and odds ratios. First, you have to explain this choice somewhere in the methods. Which pooled effect sizes are represented in the results? Is it the odds ratio?
  • Please report the NOS scoring by item and study in a supplementary file.
  • The authors also performed a pooled analysis of the risk of GBS following influenza infection. This was not among the study aims. I would suggest to remove this analysis.

Discussion

  • The phrase “In spite of incomplete strength of evidence, the pooled effect size of 1.15 (95% CI: 0.97-1.35) was conclusive, being very similar to the results of another meta-analysis published in 2015, i.e. a marginally vaccine-associated GBS risk of 1.22 (95% CI: 1.01-1.48)” should be rewritten. I would never say that your results are conclusive. Compared with a previously published SRMA your results did not reach an alpha of <0.05, by adding a few additional studies. Moreover, you have still a 15% increase that is, for example, significant at p<0.10. I strongly recommend to calculate the 95% prediction intervals in order to see in which interval the next study will lie.
  • Your results are different from all the three previously published SRMAs that found a significant association. This is why your results must be interpreted very cautiously.
  • Moreover, the subgroup analysis of the included cohort studies showed a statistically significant odds. Cohort studies have a higher grade of evidence as compared with case-control studies. All these observations should be discussed.
  • The phrase “A total of 5 studies showed a strong association of GBS with influenza, ILI or upper respiratory 243 infections commonly referred to under the umbrella term of influenza-like illness” is not correct. Laboratory-confirmed influenza, ILI and ARI are three completely different clinical entities.
  • Please describe your study limitations.

Author Response

Responses to comments of Reviewer No 2

First, I would like to thank you for your valuable comments/suggestions.

The paper must be reviewed by a native English speaker;

Answer:

The paper has been reviewed by a native English speaker.

The paper must follow the PRISMA guidelines.

Answer:

Thank you for your advice. The review was conducted in accordance with the PRISMA guidelines.

Completion/Correction (lines 53-54):

This systematic review was based on the guidelines outlined in the PRISMA Statement [27] using …

Moher D, Shamseer L, Clarke M, Ghersi D, Liberati A, Petticrew M, Shekelle P, Stewart LA; PRISMA-P Group. Preferred reporting items for systematic review and meta-analysis protocols (PRISMA-P) 2015 statement. Syst Rev. 2015 Jan 1;4:1.

The study rationale should be described better. Given that you are updating an existing SRMA, you should describe the main results of the SRMA by Martin Arias et al. Moreover, the same research group updated their SRMA in 2019, by identifying further three studies (Sanz Fadrique R, Martín Arias L, Molina-Guarneros JA, et al. Guillain-Barré syndrome and influenza vaccines: current evidence. Rev Esp Quimioter. 2019 Aug;32(4):288-295.). The latter paper was not cited here. Why it is important to update the existing SRMA?

Answer:

We did not conduct an update of both works [Martin Arias 2015, Sanz Fadrique 2019]. We tried to perform an independent meta-analysis. We noticed that the authors of the first study did not always respect the published effect size, including 95% confidence interval.

Moreover, only one published meta-analysis focused on the relationship between GBS and seasonal TIV vaccination. All the others evaluated only a monovalent vaccine used against pandemic influenza in 2009-10. That was why we decided to assess potential GBS onset after a commonly used trivalent inactivated vaccine against seasonal influenza only.

The phrase “Therefore, we conducted this meta-analysis where all currently available epidemiologic studies were entered to develop a risk estimate of influenza vaccine-associated GBS” is misleading. You cannot know whether you included all available studies. For instance, the following studies are missing: 1. Sandhu SK, Hua W, MaCurdy TE, Franks RL, Avagyan A, Kelman J, et al. Near real-time surveillance for Guillain-Barre syndrome after influenza vaccination among the Medicare population, 2010/11 to 2013/14. Vaccine. 2017;35(22):2986-92; 2. Ghaderi S, Gunnes N, Bakken IJ, Magnus P, Trogstad L, Haberg SE. Risk of Guillain-Barre syndrome after exposure to pandemic influenza A(H1N1)pdm09 vaccination or infection: a Norwegian population-based cohort study. Eur J Epidemiol. 2016;31(1):67-72.

Answer:

We agree. We were convinced that all observational studies were enrolled. You are right. The paper by Sandhu 2017 was eligible but it was not shown in our search. Conversely, the other one [Ghaderi 2016] did not evaluate TIV vaccination but only pandemic influenza vaccination. Therefore, it was not eligible for our study. We made the following correction.

Completion/Correction (lines: 44-45):

Therefore, we conducted this meta-analysis where all available epidemiological studies from our computerised search were entered to develop a risk estimate of influenza vaccine-associated GBS.

The phrase “To exclude potential risk or confounder factors such as “experimental” influenza vaccine, unpredictable circulating strain or extraordinary vaccination, we focused only on GBS occurrence after immunisation against seasonal influenza carried out every year using a common inactivated trivalent vaccine” has several issues. First, what do you mean by “experimental” vaccines?

You included only observational studies; this means that only licensed products could be investigated in primary research.

Second, you cannot exclude the confounding by “unpredictable circulating strains”: both the virus population and vaccine composition change almost each season. In order to minimize such an effect you have to pool the available studies separately by season and location and/or to perform a meta-regression analysis. By the term ”extraordinary” vaccination do you intend the pandemic one? What about quadrivalent vaccines? They have been the most widely used vaccines for the recent 2-3 years (in several high-income countries).

Answer:

Experimental vaccines were considered to be monovalent influenza vaccines specifically designed for pandemic influenza vaccination in 2009-10. We agree that the statement should have been better worded.

Completion/Correction (lines: 47-51):

We focused only on GBS occurrence after immunisation against seasonal influenza carried out every year using a common inactivated trivalent vaccine to exclude effect of extraordinary use of a monovalent pandemic influenza vaccine with or without adjuvants. We aimed to evaluate the effect size (ES) not affected by potential confounders resulting from the new pandemic strain of influenza A subtype H1N1, rapid manufacture and distribution of a pandemic influenza vaccine, etc.

Do you have a protocol? Have you registered it. If yes, you should provide a reference to it or at least attach it in a supplementary file.

Answer:

The protocol of our study was written in the original language, i.e. in Czech. It contains 58 pages and is confidential. The protocol was not registered because we could not find any registry for meta-analysis studies. It is not usual to include the protocol in a supplementary file and we do not consider it appropriate.

You have to report at least one search syntax in full.

Answer:

Search syntax was added to a supplementary file.

Did you search the grey literature? These sources are of interest here (e.g. VAERS-based evaluations).

Answer:

Our search was performed using the reported databases. If there had been some VAERS-based study as an output of search then it would have been included. No publication in the grey literature was identified.

The inclusion criterion 1 has to be reviewed. For instance, the cohort design does not have cases and controls.

Answer:

It was stated: ... observational studies with controls such as case-control (C-C) and cohort (C) studies ... We did not claim that cohort studies contain cases and controls. Cohort studies are divided into cohorts with exposure and without exposure. A cohort with no exposure is assessed as a control cohort, therefore a cohort study also belongs to controlled studies.

Why the randomized controlled studies were excluded? Sufficiently powered RCTs (usually efficacy trials) have thousands of study subjects. You may have a rate of GBS in both vaccine and placebo/non-influenza vaccines arms.

Answer:

Randomized controlled trials are clinical trials. These studies are not primarily aimed at monitoring the incidence of autoimmune diseases after vaccination. Unfortunately, their quality is burdened by the lack of knowledge of GBS before entering the study and by confirmation of GBS after vaccination. We would have included an RCT in the meta-analysis only if there had been an RCT summary focused on the occurrence of autoimmune diseases after influenza vaccination. Unfortunately, such a summary was not available.

We excluded observational studies that had a control group vaccinated with another vaccine. Such an effect size would not evaluate the impact of influenza vaccination on the GBS incidence, but it would assess whether influenza vaccination increases the incidence of GBS compared to another vaccination. This was not the objective of our work.

Please specify the exclusion criteria. The phrase “Studies without usable data were excluded” is not sufficient (it is also “unclear”).

Answer:

We agree. The statement was changed.

Completion/Correction (lines: 67-68):

…. outcome of interest measured using effect size (ES) of vaccination on GBS occurrence including 95% confidence interval (excluding studies without a reported effect size or entry for odds ratio calculation).

Please specify which effect sizes you have considered a priori.

Answer:

The effect size was general, and it was obtained from any reported effect size such as relative risk, incidence rate ratio or odds ratio. A transformation of any ratio was not performed to exclude a "transformation/statistical" bias.

Completion/Correction (lines: 83-84):

Study-reported relative risks, incidence rate ratios or odds ratios with or without their adjustment were used for pooled quantitative analysis, i.e. for pooled ESs.

The PICOS algorithm must be specified. For instance, which population? How did you categorize the population?

Answer:

It has been described that all published ES independent of gender, age at the longest time window - see 2/11 - lines 74-75 were included in the meta-analysis.

Factors of adjustment were not reported in the results.

Answer:

An adjustment cannot be performed in a meta-analysis. Sensitivity analyses were therefore performed with respect to the selected predictors - see section 3.3.

Do you have a reference for the NOS scale score categorization? As far as I know currently there is no a well-defined threshold for distinguishing between high and low quality studies.

Answer:

This was reported on page 2/11, lines 78-79

Because the cut-off of NOS score was not accurately determined, we evaluated the result on a set of quality studies ≥7 NOS and its robustness was assessed across the entire set of studies irrespective of their quality.

The list of excluded studies should be reported in a supplementary file.

Answer:

A total of 399 articles were discarded. We think that it is not interesting for readers to report them in the supplement. Moreover, no other meta-analysis study reported excluded publications specifically.

Is this a typo ≤43 and ≤43-365 days?

Answer:

A ≤42 day interval means the following period: 1-42 days

An interval of ≤43-365 days means: 1-43 days to 1-365 days. We did not find a typo error.

Completion/Correction (lines: 225-227):

Our sensitivity analysis tested the impact of shorter (i.e. interval from 1st day to 42th day) and longer (i.e. interval from 1st day to 43-365th day) post-vaccination time windows on the GBS risk estimate.

No results from the declared publication bias analysis were reported.

Answer:

The results are:

bias interception: -0.76 (95% CI: -2.10-0.56), p-value = 0.241 (no effect of small studies)

Trimming estimator: Linear

iteration |  estimate    Tn    # to trim     diff

----------+--------------------------------------

    1     |    0.150    151         0         325

    2     |    0.150    151         0           0

Note: no trimming performed; data unchanged (0 absent unpublished studies)

Absence of asymmetry displayed by confunnel plot

We believe that the statement on page 5/11 lines 170-171 is quite sufficient. However, if an asymmetry, bias interception or trimming and filling were found, the results should be presented.

No results of the GRADE evaluation were reported.

Answer:

GRADE evaluation was defined on page 2/11 –lines 89-94. In section 3.2., all items of the GRADE evaluation were listed in the text, i.e. page 5/11 - lines 161-173: limitations, inconsistency, indirectness, imprecision and publication bias.

You pooled together different study designs and associated different effect sizes like risk and odds ratios. First, you have to explain this choice somewhere in the methods. Which pooled effect sizes are represented in the results? Is it the odds ratio?

Answer:

We followed the same procedure as in other meta-analyses, i.e. the effect size was generally assessed not as a relative risk or odds ratio. Conversion of the odds ratio to relative risk or vice versa is very limited and can usually cause statistical bias. Therefore, a conservative approach was adopted to assess the effect size by including all published ratios, regardless of whether there was a relative risk or an odds ratio. This approach led to reduction of the risk of statistical bias due to inaccurate transformation. This procedure resulted in a more accurate outcome and was represented by the general effect size usually used in meta-analyses.

In addition, it allowed us to use all published ratios irrespective of their specificity.

Please report the NOS scoring by item and study in a supplementary file.

Answer:

NOS scores are reported in a supplementary file.

The authors also performed a pooled analysis of the risk of GBS following influenza infection. This was not among the study aims. I would suggest to remove this analysis.

Answer:

This analysis is only complementary to illustrate the effect of influenza / ILI on the incidence of GBS. We consider it an important point of interest for readers of Vaccines (Basel), although it was not the primary objective of our meta-analysis.

The phrase “In spite of incomplete strength of evidence, the pooled effect size of 1.15 (95% CI: 0.97-1.35) was conclusive, being very similar to the results of another meta-analysis published in 2015, i.e. a marginally vaccine-associated GBS risk of 1.22 (95% CI: 1.01-1.48)” should be rewritten. I would never say that your results are conclusive. Compared with a previously published SRMA your results did not reach an alpha of <0.05, by adding a few additional studies. Moreover, you have still a 15% increase that is, for example, significant at p<0.10. I strongly recommend to calculate the 95% prediction intervals in order to see in which interval the next study will lie.

Answer:

We agree. The statement has been modified.

All results were evaluated at a significance level of alpha = 0.05. In other words, a 5% error was accepted. The increased incidence of GBS after seasonal influenza vaccination did not reach statistical significance. If accepting an 10% error, i.e. the significance level of alpha is increased to 0.1, the ES outcome remains unchanged, i.e. ES = 1.15 (90% CI: <1.0 to 1.32), p = 0.101.

Furthermore, a prediction interval of 95% ES = 1.15 (95% PI: 0.82 to 2.23) was calculated to confirm the accuracy of the result. The calculation was conducted according to IntHout 2016.

Moreover, the prediction interval was tested with a 20% error, i.e. also 80% PI - ES = 1.15 (80% PI: 0.98 to 1.86) - led to the absence of an association between GBS and seasonal TIV vaccination.

IntHout J, Ioannidis JP, Rovers MM, Goeman JJ. Plea for routinely presenting prediction intervals in meta-analysis. BMJ Open. 2016 Jul 12;6(7):e010247.

Limitation of the prediction interval:

Limitations are that the calculations and inferences for the prediction interval are based on the normality assumption, which is difficult to ensure. Further, the interval will be imprecise if the estimates of the summary effect and the between-study heterogeneity are imprecise, for example, if they are based on only a few, small studies. Inferences based on the prediction interval are only valid for settings that are similar (exchangeable) to those on which the meta-analysis is based.

Completion/Correction (lines: 273-275):

In spite of incomplete strength of evidence, the pooled ES of 1.15 (95% CI: 0.97-1.35) was clearly very similar to the results of another meta-analysis published in 2015, i.e. a marginally vaccine-associated GBS risk of 1.22 (95% CI: 1.01-1.48)

Your results are different from all the three previously published SRMAs that found a significant association. This is why your results must be interpreted very cautiously.

Answer:

All three previously published SRMAs focused on a monovalent vaccine against pandemic influenza. Only one of them tried to evaluate also the effect size of a seasonal influenza vaccine for the purpose of comparing seasonal versus pandemic influenza vaccination. The outcome of this study was discussed. The other way round, we do not think that our results could be compared with the outcome of a completely different vaccine.

Moreover, the subgroup analysis of the included cohort studies showed a statistically significant odds. Cohort studies have a higher grade of evidence as compared with case-control studies. All these observations should be discussed.

Answer:

The identified associations were discussed and justified. The main issue was a low number of studies in analyses of sensitivity.

The phrase “A total of 5 studies showed a strong association of GBS with influenza, ILI or upper respiratory 243 infections commonly referred to under the umbrella term of influenza-like illness” is not correct. Laboratory-confirmed influenza, ILI and ARI are three completely different clinical entities.

Answer:

Only influenza-like illness or upper respiratory infection has been reported by the authors. We adopted it as ILI and described it in our article. We consider it fair to the readers and in no way misleading. None of the authors of eligible studies reported ARI or laboratory-confirmed influenza.

Please describe your study limitations.

Answer:

Such a para has been added.

Completion/Correction (lines: 337-339):

Admittedly, not all published references were identified in this systematic review and meta-analysis, albeit no additional references were retrieved from the reference lists in the articles emerging from literature search, a fact we accept it as a limitation of our study.

Round 2

Reviewer 2 Report

The manuscript has been improved. I have no further comments.